# Polarization Dependent Photoinduced Supramolecular Chirality in High-Performance Azo Materials

**DOI:** 10.3390/molecules26102842

**Published:** 2021-05-11

**Authors:** Sekvan Bagatur, Marcel Schlesag, Thomas Fuhrmann-Lieker

**Affiliations:** Physical Chemistry of Nanomaterials, Institute of Chemistry and Center for Interdisciplinary Nanostructure Science and Technology, University of Kassel, Heinrich-Plett Str. 40, 34132 Kassel, Germany; sbagatur@uni-kassel.de (S.B.); marcel.schlesag@student.uni-kassel.de (M.S.)

**Keywords:** azo materials, supramolecular chirality, polarimetry, azo polymers, polarization lithographie, polarimetry

## Abstract

Here, we will show photo-induced supramolecular chirality in thin films of achiral amorphous polymers with azo groups in their side-chain. A matter of particular interest is the effect of various film thicknesses on azimuthal rotation and ellipticity of incident/transmitted polarized light. Furthermore, we investigated the temporal stability of inscribed chirality. By polarimetric measurements, we found out that the azimuthal rotation gets higher with layer thickness. In this scope, we were able to measure a very high azimuthal rotation of $\Delta \psi /d=112.5^{\circ }$/μm. The inscribed chirality was stable for several days. Furthermore, we investigated the time-resolved behavior of incident and transmitted polarization ellipticities for various thicknesses. The time dependency may be explained by a two-step process: (1) fast *trans-cis*-isomerization resulting in photo-orientation and (2) slow photo-induced mass flow.

## 1. Introduction

Azo materials as light-responsive materials are under investigation for decades in countless experiments. Since the discovery of its photomechanical property the interest of researchers all around the world increased significantly. Undoubtedly, the remarkable work of H. Rau [1] regarding the classification of azo materials was an important contribution for research progress. Particularly the pioneering work of Rochon et al. [2] as well as Kim et al. [3] regarding holographic surface relief grating was an important contribution. In the recent years novel phenomenon were discovered: photo-induced stretching of colloids [4,5,6,7,8,9,10] and micro-particles, photo-induced self-folding of 2D films [11], photo-induced particle movement [12] are just few examples.

The characteristic of azo materials is a polarization dependent photo-isomerization upon irradiation (see Figure 1). Here, the probability *W* of *trans-cis*-isomerization depends on the angle α between polarization axis *P* of incident light and transition dipole moment *M* of the azo molecule: $W\propto \cos^{2}\alpha$.

Since the absorption bands of *trans* and *cis*-configuration overlap, isomerization is repeated continuously during illumination. After each cycle, the azo molecule rotates slightly until *M* is perpendicular to *P*. Hence, the probability amounts to zero and no photo-isomerization occurs. This process is called photo-orientation [13,14]. For side-chain azo polymers it means that the azo containing side-chain undergoes the process of photo-orientation, while the position of the backbone is changed slightly [14].

Inscribed structures are highly dependent on incident light polarization. A very interesting effect is the photo-induced introduction of circular birefringence [15] and circular dichroism in liquid crystalline azo polymers using circularly polarized light (CPL) [15]. The same manipulation of optical properties was additionally performed in amorphous and liquid crystalline azo films (thickness ≈5μm) using elliptically polarized light (EPL) [16]. Later Kim et al. [17,18] and Sumimura et al. [19] introduced optical activity into amorphous azo layers using EPL. However, no chirality was measured using an ellipticity (ϵ) close to complete CPL [18]. Whereas, Cipparrone et al. have shown optical rotation (Δψ) due to photo-induced supramolecular chirality (PSC) using almost complete CPL on a film of a side-chain azo copolymer [20]. The direction of optical rotation depends highly on the handedness of incident polarized light: left-handed circular polarization results in left-handed optical rotation and right-handed circular polarization leads to right-handed optical rotation [13,15,16,17,18,19,21,22,23]. Moreover, a helical arrangement of chromophores after irradiation with EPL is assumed [13,16,17,18,19]. Nikolova et al. assume a similar arrangement as cholesteric liquid crystals with high pitch *p*. However, since the mechanism is not fully understood, it is still not clear whether or not a helical arrangement from the amorphous phase is really occuring. It seems assured that the photo-induced rearrangement does not only happen on the film surface but also in bulk material. This would imply that change of optical rotation is thickness-dependent. The work of Mazaheri et al. [24] can be interpreted as underlining the conception of rearrangement in bulk material, when optical rotation varies for different layer thicknesses. Therefore, it corroborates the statement of structural reorientation from amorphous phase to cholesteric-like arrangement.

Here, we report about PSC using *l*-EPL on high-performance azo polymers, which are known for their high optical birefringence [25,26,27]. For this purpose, we investigate the temporal progression of azimuthal rotation upon exposure and its dependence on film thickness. An exemplary comparison of literature values for the azimuthal rotation in relation to its layer thickness *d* shows the following measured values: $\Delta \psi /d\;=\;6^{\circ }/\mu m$ [15], $\Delta \psi /d=8.33^{\circ }/\mu m$ [20], $\Delta \psi /d=10^{\circ }/\mu m$ [18], $\Delta \psi /d=41^{\circ }/\mu m$ [19]. Recently, we measured a very high azimuthal rotation with a value of $\Delta \psi =59^{\circ }$ [28]. In relation to its film thickness, the rotation amounts $\Delta \psi /d=66.6^{\circ }/\mu m$. In this work, we almost doubled the azimuthal rotation from our previous work by adjusting the incident ellipticity and measured a maximum value of $\Delta \psi /d=112.5^{\circ }/\mu m$. To our knowledge, this is the highest measured azimuthal rotation in amorphous PMMA-backboned side-chain azo polymers so far.

Furthermore, we investigate the effect of long-term exposure (700min) and the stability of introduced chirality over several days. Compared to incident ellipticity of *l*-EPL, we observe that the transmitted ellipticity is changed for all time and thickness-dependent measurements in this work.

Based on our previous research and literature, we can assume that light-treatment with EPL results in supramolecular chirality within the irradiated spot on the sample (Figure 2). Therefore, the azimuthal angle $\psi_{a}$ of analysis beam rotates while traversing that spot (Figure 2a). This is quite similar to optical activity of a cholesteric liquid crystal. We can imagine this as twisted layers of azo groups with the same mean orientation (Figure 2b)—comparable with a director in liquid crystals. The double-headed arrows in Figure 2b visualize the mean orientation of each layer. Conceivable is that the transition between each layer is continuous.

As seen in Figure 2a, an ellipse is described by its major axis *a* and minor axis *b*. The “roundness” is determined by the ellipticity
(1)$$\epsilon =\arctan\left(\frac{b}{a}\right).$$

The angle between the orientation of the major axis in a coordinate system where light propagation is along *z*-axis and the *x*-axis is defined as the azimuthal angle ψ. After transmission through a sample with PSC, the azimuthal angle rotates at a certain angle (ascribed as azimuthal rotation Δψ). Additionally, the transmitted ellipticity can differ from incident condition, which is equal to a change in intensity of major and minor axis (Figure 2a).

In this work, we exclusively used *l*-EPL for our experiments. The polarization handedness was determined by using the same rotational direction, as it would take to rotate the incident linearly polarized light (LPL) at the smallest angle to be aligned (parallel) according to slow axis of the quarterplate [29] (p. 370). Caution should be taken here: whenever we talk on handedness of circularly or elliptically polarized light, we mean that handedness while looking opposite to light propagation as an observer.

## 2. Materials and Methods

For inducing PSC we have used PMMA-backboned side-chain azo polymers chosen as representatives of two different classes: Polymer **2** is an amorphous three-core homopolymer and Polymer **3** is a mixed two-core and three-core amorphous copolymer (see Figure 3a and Table 1). These polymers are characterized by their extend conjugated π-electron system and the push-pull substitution by amine and (bi)cyano groups. Since we are interested in reaching high azimuthal rotations, Polymer **P1** as used in [28] with a nitro-substituent was excluded because of its low values for azimuthal rotation. This behavior is explainable by its comparatively lower polarizability due to the smaller two-core π-electron system. The polymers were synthesized by radical polymerization of monomeric methacrylates [25,26,27].

The raw materials have been dissolved in THF (Uvasol^®^-quality) with different concentrations (20, 40 or 80 mg/mL) and stirred at ambient temperature for several days. A diluted fraction of the solutions were used to acquire absorption spectra using a Lambda 900 UV/VIS/NIR Spectrometer (PerkinElmer Instruments, Waltham, MA, USA). The spectra are seen in Figure 3b. The solution was used for spin-coating on glass substrates, which have been cut out of object slides. A self-made spin-coater at speeds between 500–800 RPM was used. In order to eliminate remaining solvent, the samples were heated up to 80 °C for 1 h. It should be taken into account that the polymers were prepared to be in the amorphous phase, i.e., the glassy phase got frozen upon spin-coating from solution. Thicknesses were measured with a spectroscopic ellipsometer (*VASE^®^*, J.A. Woollam Co., Inc., Lincoln, NE, USA). Afterwards, the samples were stored in darkness for at least 3 weeks to assure having all azo molecules in the thermodynamically stable *trans-*configuration. Light treatment for inducing PSC was performed using a continuous wave DPSS-laser (diode pumped solid state laser) at the wavelength of $\lambda_{e}=473\;\mathrm{nm}$, which is in the absorption of the three-core side-chain and at the edge of the two-core azo polymer (Figure 3b). Here, the subscripts ‘e’ denote the excitation beam. Consequently, either absorption band are addressed at different degrees: The two-core side-chain absorbs quantitatively a higher intensity than the three-core side-chain. The laser beam (fluence of ≈$1200\;\mathrm{mW}/{\mathrm{cm}}^{2}$) was left-handed elliptically polarized by using a linear polarization filter (Edmund Optics, Mainz, Germany) and a quarter-wave plate (WPQ10M-473, Thorlabs, New Jersey, USA) at various angles (see Table 2). For time-resolved development of PSC, the samples were irradiated for 2 s, 5 s, 10 s, 20 s, 30 s, 1 min, 3 min, 5 min, 10 min, 30 min, and 70 min on exactly the same spot. Long-term experiments comprise additional irradiation times at 70 min-steps up to 700 min. Between each irradiation with blue laser polarimetric measurements were performed with a red laser at $\lambda_{a}=633\;\mathrm{nm}$ (subscript ‘a’: analysis beam) outside absorption (see Figure 3 and Figure 4). The beam of either laser was set to be left-handed elliptically polarized (*l*-EPL).

While PSC is introduced on the sample by blue laser, the red laser is blocked by the shutter r-SH (Figure 4a,b). After light-treatment, the shutter of blue laser b-SH was closed to block the beam and the sample S was rotated at $90^{\circ }$ counterclockwise (*x*-*z*-plane) for the polarimetric measurement. Here, the r-SH was opened and red laser beam traverses through S at exactly the same spot as the blue laser beam (Figure 4c). The self-made polarimeter comprises a HeNe-Laser, a polarizer (Edmund Optics, Mainz, Germany), a quarter-wave plate depending on the requirement, an *x*-*y*-movable sample holder on a rotation stage (OWIS DMT 100, Staufen i. Br., Germany), an analyzer (Edmund Optics, Germany) mounted on a compact rotation stage (OWIS DMT 65, Staufen i. Br., Germany). Polarimetric measurements were performed by rotating A around the red laser-beam (*z*-axis of analysis laser) with a resolution of $0.1^{\circ }$. The photodiode PD, connected to a multimeter, detects the intensity for each angle of A. Measured angles with its corresponding intensities were recorded by a multimeter (Agilent U3606A, Agilent Technologies, Santa Clara, CA, USA) as voltage signal. This procedure were done for various irradiation times $t_{e}$. All experiments were done by irradiating the samples from either, the azo-side (front) and glass-side (back). A detailed scheme of the experimental setup can be seen in Figure 4.

A self-written LabVIEW program was used for controlling the shutters, rotational stages of the sample and analyzer as well as for recording intensity data from the multimeter with its corresponding angle of the analyzer. An Arduino Mega 2560 was connected between the movable/rotatable components and the computer for control of corresponding stepper motors (marked with * in Figure 4). The multimeter was directly connected to the computer.

All following calculations and data analysis were done with self-written programs using Python 3.8. Here, all measurements were smoothed with the Savitzky-Golay method with a window size of 201 (out of 3601 data points for one polarimetric measurement) and polynomial order of 5. Uncertainties for any azimuth angle $\delta \psi_{n}$ were calculated as the sum of mean absolute deviation of each polarimetric measurement and accuracy of reading (one scale unit). For the intensity, accuracy of reading were calculated using the mean distance between each intensity data within one polarimetric measurement. Since the data had some noise, an estimated uncertainty of 10% was added. By using this, accuracy for ellipticity was calculated using gaussian method of error propagation.

## 3. Results

Firstly, we will show and discuss our time-resolved polarimetric measurements done with HeNe-laser ($\lambda_{a}=633\;\mathrm{nm}$) in dependence on layer thickness. The first measurement of this series is a polarimetric measurement of a non-treated sample. Afterwards exactly the same spot was irradiated with DPSS-laser at $\lambda_{e}=473\;\mathrm{nm}$ for various exposure times *t*. Between each exposure time, a polarimetric measurement was done in order to measure azimuthal angle $\psi_{a,t}$ and transmitted ellipticity $\epsilon_{a,t}$ (subscript “t” refers to the characteristics of transmitted light; “*t*” to exposure time). The azimuthal rotation Δψ is obtained by calculating the difference of each transmitted $\psi_{t}$ for its corresponding exposure time *t* to the initial, transmitted azimuthal angle $\psi_{a,t}\left(t=0\;s\right)$ for the non-treated area. Consequently, azimuthal angles $\psi_{a,t}\left(t=2\;s–70\;\min\right)$ of transmitted beam from all polarimetric measurements were compared to the reference value $\psi_{a,t}\left(t=0\;s\right)$. In order to examine the stability of PSC, long-term measurements of 48 h and further 72 h were done without excitation.

The change of transmitted ellipticities of analysis laser beam $\epsilon_{a,t}$ for each polarimetric measurements were compared with incident $\epsilon_{a,i}$ (subscript “i” refers to incident beam). Ellipticities for the same layer thicknesses were calculated with the measured intensity at its maximum and minimum values using Equation (Equation 1). The quarter-wave plates were set on the same angle for either laser resulting in *l*-EPL. However, as seen in Table 2, the calculated ellipticities based on measured intensities differ for the excitation laser $\epsilon_{e}$ and for the analysis laser $\epsilon_{a,i}$. The azo layers were irradiated at ellipticities of $\epsilon_{e,t}=3.54^{\circ },9.11^{\circ },18.01^{\circ }$ and $\epsilon_{a,i}=10.90^{\circ },19.89^{\circ },31.15^{\circ }$ (see Table 2). The high ellipticity of $\epsilon_{a,i}=31.15^{\circ }$ were chosen based on the work of Nikolova et al. [16,30]. According that, the maximum azimuthal rotation for amorphous azo polymers can be achieved with an ellipticity around $\epsilon_{i}\approx 30^{\circ }$.

For all following measurements, the combination of $\epsilon_{e,i}$ and $\epsilon_{a,i}$, namely ϵ(1), ϵ(2), ϵ(3) was kept for all measurements.

### 3.1. Polarimetric Measurements: Dependency on Exposure Time and Layer Thickness

Layer thicknesses in ≈100nm intervals (**P2**: ≈80,190, and 250nm; **P3**: ≈76,175, and 270nm) were chosen in order to investigate the dependency on Δψ. For covering a wide range of film thicknesses, additional very low and very high thicknesses were chosen (**P2**: ≈36 and 721nm; **P3**: ≈38, and 775nm). With the very high thicknesses it is aimed to achieve high azimuthal rotations.

For all measurements it was observable that the maximum azimuthal rotation is achieved very fast. The tendency is that thin layers reach the maximum after an exposure time of up to 10 min. For the thickest layers it takes up to 30 min (see Figure 5, left column). However, the rotation decreases afterwards for the most measurements at different rates. As seen in Figure 5 and Figure 6d–f, values for maximum azimuthal rotation seems to increase roughly linearly. The lowest value of $\Delta \psi_{\max}=1.2^{\circ }$ for the sample with d=38nm represents the minimum reliable and significantly verifiable azimuthal rotation in this work. The sample d=36nm, **P2** shows slightly higher minimum values of $\Delta \psi_{\max}=2.3–2.6^{\circ }$. Furthermore, the measurements show that higher maximum values of azimuthal rotation $\Delta \psi_{\max}$ are obtained with increasing layer thickness and incident ellipticity of excitation laser beam $\epsilon_{e,i}$. For **P2** we were able to measure the highest value of $\Delta \psi_{\max}=81.1^{\circ }$. Referred to its layer thickness, the azimuthal rotation yields a value of Δψ/d=112.5${}^{\circ }$/μm. To our knowledge, this is the highest measured value for photo-induced optical rotation in amorphous side-chain azo polymers. For **P3** we reached a maximum azimuthal rotation of Δψ/d=97.5${}^{\circ }$/μm. Observable are higher rotations for irradiating the sample from the azo-side.

In summary, four main tendencies are observable: (1) Increase of Δψ occurs very fast for thin layers and slower for thick layers. (2) The thicker the layer, the higher $\Delta \psi_{\max}$. (3) The higher the incident ellipticity $\epsilon_{e,i}$, the higher $\Delta \psi_{\max}$. However, further increasing of $\epsilon_{e,i}$ would probably again decrease the value of $\Delta \psi_{\max}$ because of decrease in anisotropic characteristics of incident light as shown by Nikolova and Kim et al. [16,17,18,30] (4) Azimuthal rotation decreases slowly after reaching its maximum value. The rate of decrease differs for different thicknesses and incident ellipticities.

The long-term excitation with $\lambda_{e}$ for exposure times up to 700 min shows for all samples a decrease of azimuthal rotation after reaching its maximum value (Figure 7). This is independent on the irradiated side of the sample (azo or glass-side).

After $\Delta \psi_{\max}$ is reached very fast, the decay is very fast at the beginning and slowly went over to an equilibrium state. Using $\Delta \psi_{\max}$ as the reference, the decay is about 5–33% for thicker layers, while higher relative decay is observable for **P3** (see Table 3). For high incident ellipticities, the relative decay is decreased for either material.

The samples used in Figure 7a,b had a thickness of 721 nm and 775 nm for all ellipticity combinations ϵ(n). The thin films of polymer **P2** used for polarimetric measurements in Figure 7c had a thickness of 77 nm for ϵ(1) and ϵ(2), whereas for ϵ(3) a film thickness of d=82nm) was used. In case of polymer **P3** a sample with d=76nm were used for all three combinations of incident ellipticity (see Figure 7d).

### 3.2. Temporal Stability of PSC

For investigation of the temporal stability of inscribed PSC, exemplary two samples with high thicknesses were used for each polymer (see Figure 8).For polymer **P2** and **P3**, samples with d=749nm and d=789nm were used, respectively. After the long-term excitation of 700 min with $\epsilon_{e,i}=9.11^{\circ }$, only polarimetric measurements (1h-steps for 48 h and 6 h-steps for further 72 h) were done with $\epsilon_{a,i}=19.89^{\circ }$. This is the ellipticity-combination ϵ(2) from Table 2. For stability measurements, the azimuthal angle $\psi_{a,t}$ of transmitted EPL from last polarimetric measurement during excitation was used as the reference. While **P2** shows no significant decrease of azimuthal rotation, **P3** does with a value of about $\Delta \psi =-3^{\circ }$. It seems like that either polymer has different abilities to form more or less stable structures. However, compared to $\Delta \psi_{\max}$ for each polymer, the decrease is very low. The up and down behavior after first 48 h (during measurements of 6 h-steps), may attributed to temperature differences at day and night. Furthermore, it is observable that the transmitted ellipticity increases slightly with time.

### 3.3. Behavior of Ellipticity

It is worth it to take a closer look at the ellipticity after transmission $\epsilon_{a,t}$. In accordance to literature [17,18,30,31], our measurements show a change of transmitted ellipticity $\epsilon_{a,t}$ compared to incident laser beam $\epsilon_{a,i}$. It is observable that $\epsilon_{a,t}$ is slightly decreased for the non-treated azo layers. Conceivable is a small contribution of initial anisotropy caused by spin-coating (see Material and Methods). In Figure 9 and Figure 10, the results for temporal change of $\epsilon_{a,t}$ are shown. The plotted data visualize the change of transmitted ellipticity compared to incident ellipticity of analysis beam: $\Delta \epsilon \;=\;\epsilon_{a,t}$ – $\epsilon_{a,i}$. A decrease is achieved, if Δϵ<0 and an increase, if Δϵ>0. The ellipticities were calculated using Equation (Equation 1).

For polymer **P2** films, change of ellipticities generally increases within the first 30 s of excitation and decreases after reaching a maximum (Figure 9a–c), if thickness is above d=80nm. The increase is very fast, whereas the decrease is much slower. The thickest sample shows a dominant behavior of increase at the beginning, following by a decrease of Δϵ. At highest incident ellipticity, the change of ellipticity is increasing for all films (Figure 9c). This means that the transmitted ellipticity gets more roundish. For the thinnest films with d<80nm, the beginning increase may be negligibly small and only the followed decrease, e.g. thinning of $\epsilon_{a,t}$, can be seen. Qualitatively, the change of ellipticity is independent on irradiation side, but quantitatively it is.

From smaller values of $\epsilon_{a,i}$ and $\epsilon_{e,i}$ to higher, the same development of Δϵ occurs for polymer **P3** (Figure 10). Particularly for films below d=262nm, a decrease at smaller $\epsilon_{a,i}$ and increase at higher $\epsilon_{a,i}$ is observable within the first 30 s of excitation. However, the thickest film of d=775nm shows for all incident ellipticity combinations a beginning increase and decrease after reaching the maximum value. Similarly to **P2**, the maximum value is reached very fast following by a decrease of Δϵ.

## 4. Discussion

The polarimetric measurements show very high azimuthal rotation of maximum $\Delta \psi /d\;=\;112.5^{\circ }/\mu m$ and $\Delta \psi /d\;=\;97.5^{\circ }/\mu m$ for polymer **P2** and **P3**, respectively. These values are explainable by their high polarizability, large permanent dipole moment and high birefringence. Substituted cyano-groups and introduction of a three-core side-chain result in larger anisotropy of polarizability and permanent dipole moment. Based on this, it can be concluded that areas irradiated with excitation laser achieve a higher degree of molecular order [25,26,27]. This explains the significant high optical rotations of polymers **P2** and **P3** (cf. [28]). The comparison between homopolymer **P2** and copolymer **P3** shows that the bicyano substitution at the three-core side-chain leads to more advantageous behavior for PSC than the monocyano substitution. Hence, **P2** exhibits higher azimuthal rotation than **P3**. However, the optical rotation is not only affected by the material itself but by the side of irradiation, too. The general tendency is that irradiating the sample from film side leads to higher Δψ than from the glass substrate side. It seems that anisotropic surface structures, which can be imagined as 2D-chiral surfaces (see [24]), contribute to azimuthal rotation. The difference between irradiating azo and glass-side is discussed more detailed in [28].

We can imagine PSC as a two-step process: Step (1) comprises a fast *trans-cis* photoisomerization leading to rotation of azo molecules perpendicular to incident polarization plane of excitation beam. The process of photo-orientation occurs until incident plane polarization and transition dipole moment of irradiated molecules are perpendicular to each other [13]. This would explain the fast increase of maximum azimuthal rotation $\Delta \psi_{\max}$ and change of ellipticity Δϵ. For very thin films, it is assumed that the beginning increase of Δϵ is negligibly small due to very fast photo-isomerization at step (1) and therefore not observable. In case of higher incident ellipticities, the transmitted ellipticity increases only for thinner films. This can be explained by weakened photo-orientation due to comparatively more dominant intensity of minor axis of EPL, resulting in less degree of rotation for each azo containing side-chain. Measurements with higher time resolution are needed to measure this effect. Step (2): After the fast increase, $\Delta \psi_{\max}$ and Δϵ decrease slowly until reaching an equilibrium state. The second step comprises photo-induced and polarization-dependent anisotropic mass flow of azo material. Owed to the big size of polymers and therefore its rigidness this process is slow. Since mass flow is directed from areas irradiated with higher intensity to lower intensity, the density of rotated molecules at highest beam intensity decreases. For a gaussian beam as used in this work, mass flow occurs radially from center to the edge of the beam. Consequently less amount of azo molecules are available for contribution to optical rotation, resulting in decrease of azimuthal rotation. The additional decrease after switching excitation light off is caused by *cis - trans* relaxation.

Even though the mechanism and the resulted molecular rearrangement is not completely understood so far, it seems possible that a cholesteric-like phase might be inscribed (cf. Nikolova et al. [16,30,32]). The cholesteric phase is described as oriented molecules, whereas each layer in bulk has another mean orientation of molecules (Figure 2). This is comparable with step (1). The following mass flow in step (2) can be described as anisotropic stretch, changing its direction layer by layer (twisted, stretched layers). As already shown in [4,5,6,7,8,9,10], irradiation of micro-particles consisting of azo materials with polarized laser beam leads to highly dominant stretching. Photo-induced stretching is explainable by decrease of volumetric density caused by anisotropic mass flow. A similar effect might occur in a thin, continuous film, whereas those azo molecules closest to the surface (exposed to highest intensity of light) form an anisotropic layer and rotate the polarization axis of an incident beam. Let us call this layer the “initial layer”. After rearrangement of the initial layer, the rotated beam can now stretch the layer below it, forming a second anisotropic layer, while the transition between each layer is continuous. Upon deeper penetration of light through an azo film, this procedure could be repeated, resulting in creation of further anisotropic layers. Since the intensity decreases exponentially through the film, both, photo-orientation as the first step of rearrangement and “stretched layer effect” as the second step, is supposed to weaken with increasing penetration depth. This means that layers close to surface are stronger photo-oriented and stretched than deeper layers. Consequently, optical rotation at first layers should be higher than at deeper ones during step (1) and weaken upon further irradiation in step (2). As the intensity decreases exponentially, it is possible that the optical rotation for each layer, beginning with initial layer, increases and decreases on the same way. Furthermore, the “stretched layer effect” should be erasable by rotating the sample at 90${}^{\circ }$ and repeating irradiation with exactly the same characteristics of incident beam (cf. [9,33,34]). A good sign for the “stretched layer effect” are experiments on photo-induced wrinkling. Surface structures as shown in [28] might be the result of photo-induced wrinkling as comparatively done in [35,36] (cf. surface grating). While the surface tension at the irradiated spot decreases upon photo-induced softening, followed by decrease of viscosity, the non-irradiated surrounding keeps its mechanical condition. This can create anisotropic disordered wrinkles.

In summary, a combination of both, “cholesteric-like rearrangement” and “stretched layer effect” are in accordance to our measurements with the very fast increase of $\Delta \psi_{\max}$ as well as Δϵ and their slow decrease after reaching a maximum. For both, “cholesteric-like rearrangement” and/or “stretched layer effect”, azimuthal rotation is supposed to be dependent on layer thickness and incident ellipticity. Our experiments show these dependencies and might contribute to a better understanding for photo-induced supramolecular chirality.

## Figures and Tables

**Figure 1 molecules-26-02842-f001:**
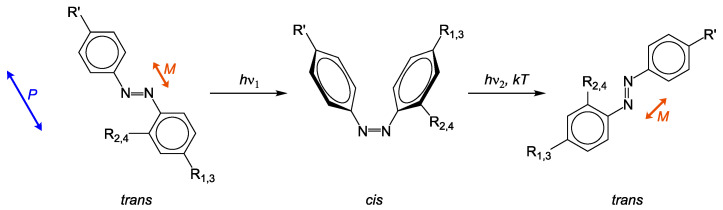
Photo-isomerization of azo chromophores. Light can not be absorbed, if polarization *P* plane is perpendicular to transition dipole moment *M*.

**Figure 2 molecules-26-02842-f002:**
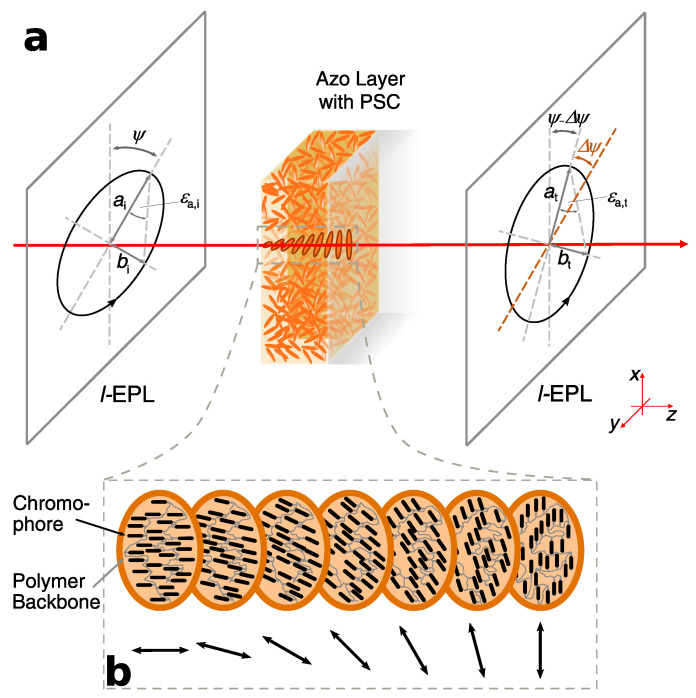
Schematical visualization of optical rotation and a possible explanation for molecular rearrangement upon irradiation with excitation beam. (**a**) Polarized light with the parameters $\psi_{a,i}$, $\epsilon_{a,i}$, $a_{a,i}$, and $b_{a,i}$. The subscript ascribes characteristics of: “a”: analysis beam; “i”: incident beam; “t”: transmitted beam. (**b**) A possible visualization of molecular rearrangement causing supramolecular chirality. Double-headed arrows refer to the mean orientation of azo molecules in each layer. This figure shows PSC and the resulted optical rotation only schematically.

**Figure 3 molecules-26-02842-f003:**
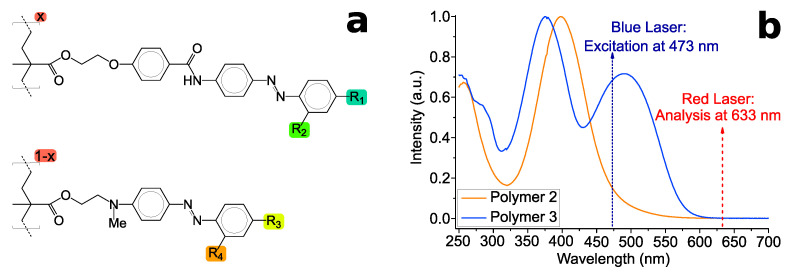
(**a**) Structural formula of PMMA-based side-chain azo polymers used in this work. *x* denotes the relative percentage of each side-chain. (**b**) Absorption spectra of used polymers. The wavelengths for excitation and analysis are labelled.

**Figure 4 molecules-26-02842-f004:**
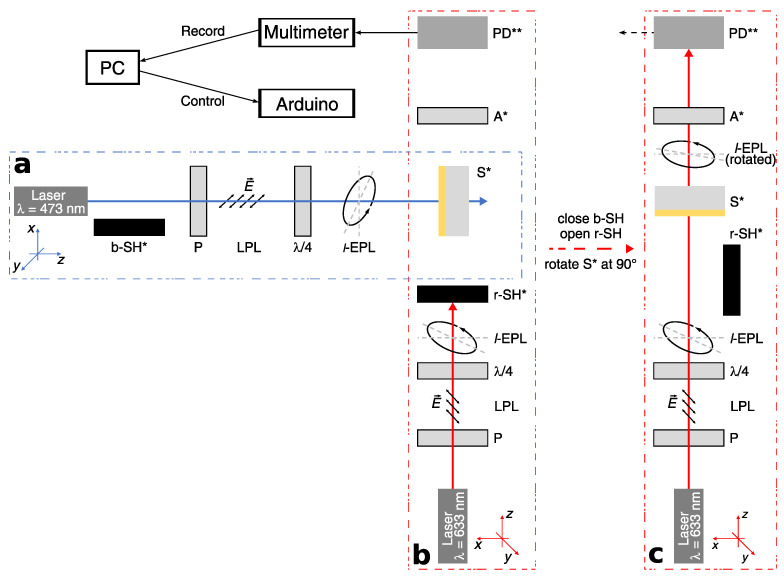
Experimental setup for inscribing PSC via a DPSS-laser and polarimetry via HeNe-laser: (**a**) The incident laser beam on the sample S* is (*l*-EPL) by using a Polarizer P and quarter-wave plate (λ/4). (**b**) The polarimetric setup is perpendicular to the inscription setup with the sample at the center. (**c**) Polarimetric measurements were performed with the red laser at (*l*-EPL). The Photodiode PD** detects the transmitted intensity, which is displayed on a multimeter. Components marked with * are connected to a microcontroller (Arduino). The mark ** means a connection to the multimeter. b-SH*: shutter for blue laser, P: Polarizer, LPL: linear polarized light, λ/4: quarter-wave plate, *l*-EPL: left-handed elliptically polarized light, S*: Sample, r-SH*: shutter for red laser, A*: Analyzer, PD**: Photodiode, $\vec{E}$: electric-field vector

**Figure 5 molecules-26-02842-f005:**
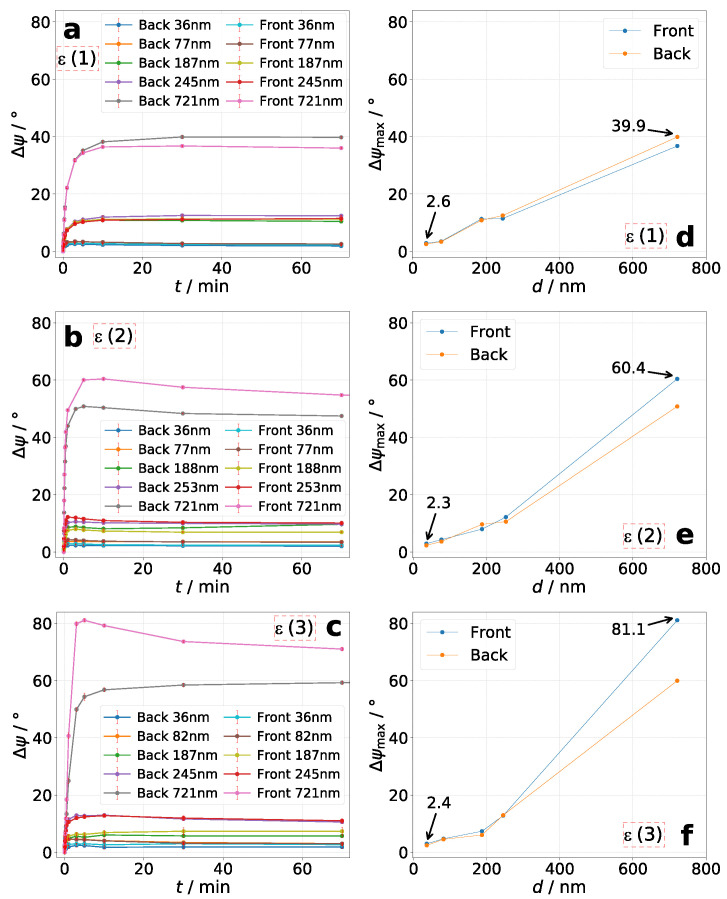
Polymer **P2**: Time-resolved azimuthal rotation Δψ for various *d* (**a**–**c**) and its corresponding maximum azimuthal rotation $\Delta \psi_{\max}$ (**d**–**f**) for each sample. Ellipticity combination of excitation and analysis laser are: (**a**,**d**): **ϵ(1)**, (**b**,**e**): **ϵ(2)**, and (**c**,**f**): **ϵ(3)** (see Table 2).

**Figure 6 molecules-26-02842-f006:**
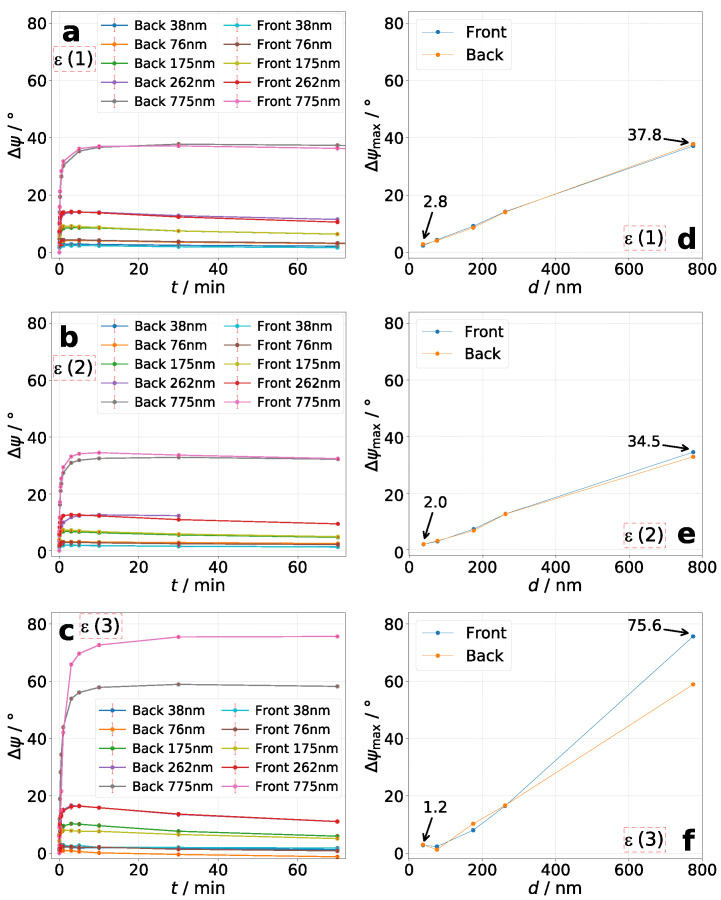
Polymer **P3**: Time-resolved Δψ for various *d* (**a**–**c**) and its corresponding $\Delta \psi_{\max}$ (**d**–**f**) for each sample. Ellipticity combination: (**a**,**d**): **ϵ(1)**, (**b**,**e**): **ϵ(2)**, and (**c**,**f**): **ϵ(3)** (see Table 2).

**Figure 7 molecules-26-02842-f007:**
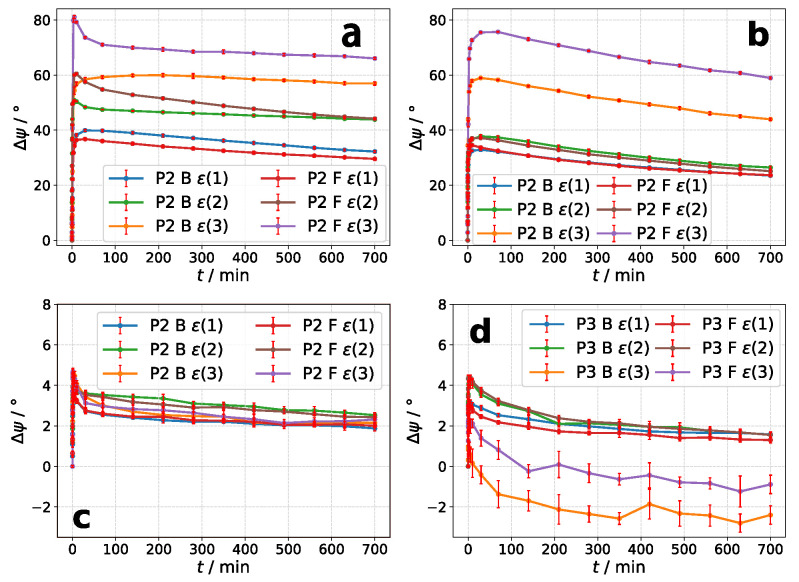
Long-term polarimetric measurements for a total exposure time of 700 min. (**a**,**c**): **P2**. (**b**,**d**): **P2**. “B” means Back for irradiating sample from glass-side, “F” means Front for irradiating from azo-side. ϵ(n) denotes the ellipticity combination of excitation and analysis laser according to Table 2.

**Figure 8 molecules-26-02842-f008:**
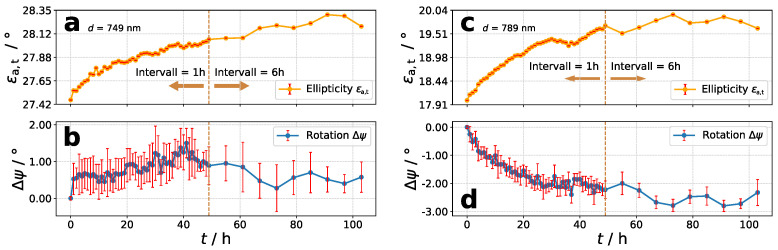
Polarimetric measurements with the ellipticity-combination of ϵ(2) for 48 h in 1 h-steps and further 72 h in 6h-steps. (**a**,**b**): Polymer **P2**, d=749nm; (**c**,**d**): Polymer **P3**, d=789nm

**Figure 9 molecules-26-02842-f009:**
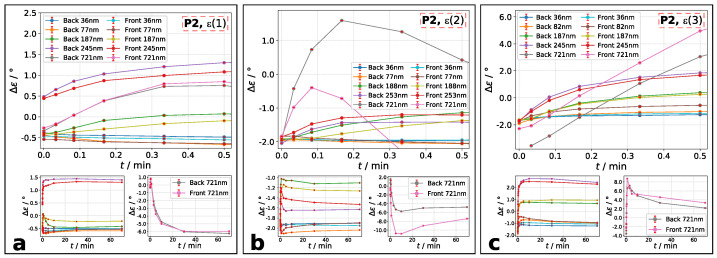
Polymer **P2**: change of ellipticities compared to incident ellipticity $\Delta \epsilon =\epsilon_{a,t}-\epsilon_{a,i}$ during the first 30 s of irradiation. Ellipticity combination is: (**a**): **ϵ(1)**, (**b**): **ϵ(2)**, and (**c**): **ϵ(3)**. Denotation: main plot: Δϵ at exposure time up to 30 s; left small plot: Δϵ for films with up to d=245nm. right small plot: Δϵ for d=721nm. The combination of ϵ(n) is labelled in each main plot.

**Figure 10 molecules-26-02842-f010:**
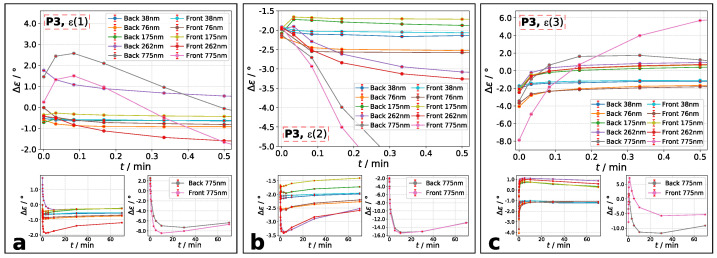
Polymer **P3**: $\Delta \epsilon =\epsilon_{a,t}-\epsilon_{a,i}$ during the first 30 s. Ellipticity combination is: (**a**): **ϵ(1)**, (**b**): **ϵ(2)**, and (**c**): **ϵ(3)**. Denotation: main plot: Δϵ at exposure time up to 30 s; left small plot: Δϵ for films with up to d=262nm. right small plot: Δϵ for d=775nm. The combination of ϵ(n) is labelled in each main plot.

**Table 1 molecules-26-02842-t001:** Functional groups of the given side-chain azo (co-)polymers used in this work. Glass transition temperatures $T_{g}$ and the relative percentage of each side-chain are listed.

Polymer	$T_{g}\;{[}^{\circ }C]$	Three Cores		Two Cores		*x* [%]	Comments
		$R_{1}$	$R_{2}$	$R_{3}$	$R_{4}$		
**2**	147	CN	CN	-	-	100	Homopolymer
**3**	116	CN	H	CN	CN	60	Copolymer

**Table 2 molecules-26-02842-t002:** Set and calculated ellipticities of incident laser beams for either lasers. Subscripts: e: excitation, a: analysis, i: incident.

Combination of Ellipticities		ϵ(1)/${}^{\circ }$	ϵ(2)/${}^{\circ }$	ϵ(3)/${}^{\circ }$
**Set on λ/4-plate**	$\epsilon_{e,i}$	15	22.5	30
	$\epsilon_{a,i}$			
**Calculated:**	$\epsilon_{e,i}$	3.54±0.08	9.11±0.08	18.01±0.09
$\epsilon_{i}=\arctan b_{i}/a_{i}$	$\epsilon_{a,i}$	10.90±0.05	19.89±0.02	31.16±0.02

**Table 3 molecules-26-02842-t003:** Decay from $\Delta \psi_{\max}$ to last value of $\Delta \psi_{t=700\;\min}$. For ellipticity combinations see Table 2.

	*d*/nm	ϵ(n)	Side	Decay		*d*/nm	ϵ(n)	Side	Decay
**P2**	721	ϵ(1)	Front	20%	**P3**	775	ϵ(1)	Front	32%
			Back	19%				Back	29%
		ϵ(2)	Front	27%			ϵ(2)	Front	33%
			Back	14%				Back	30%
		ϵ(3)	Front	19%			ϵ(3)	Front	22%
			Back	5%				Back	25%

## Data Availability

The datasets used and/or analyzed during the current study are available from the corresponding author on reasonable request.

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
