# Peer review of "Polarization Dependent Photoinduced Supramolecular Chirality in High-Performance Azo Materials"

_molecules, 2021, doi:10.3390/molecules26102842_

Round 1

Reviewer 1 Report

In this paper, the authors report the results of the dependence of red polarization state change on polarization, irradiation time, and film thickness in order to clarify the supramolecular chirality generation phenomenon induced by blue polarized light irradiation to azobenzene polymer. However, in order to get closer to elucidating the mechanism, it would be better if the following information is also presented.

1) The compounds used should be carefully explained. How did you obtain the polymers 2 and 3? If they were synthesized, please explain the synthesis method and characterization. It would be better if there is a discussion on why polymers 2 and 3 are considered to show good properties instead of polymer 1.

2) The blue light excitation of homopolymer 2 excites near the absorption edge of the azo chromophore. In contrast, the blue light excitation of the copolymer polymer 3 excites mainly the longer wavelength absorption of the two absorption peaks, according to Fig. 2b. Which of the two types of azo chromophores is being excited primarily? Also, it seems that the trans-cis photoisomerization behavior and photo-rearrangement behavior of those azo chromophores, and even mass transfer behavior would be very different depending on the situation. Please explain the experimental results regarding those.

3) The authors have discussed the dependence of film thickness and penetration depth, but please show the results of absorption spectra of the films of different thicknesses. In particular, the absorbance at 473 nm for excitation is important information for further discussion.

4) The difference between surface and backside pumping should be discussed.

5) Please show the information about the intensity of the excitation laser.

Author Response

Dear Reviewer,

thank you very much for your constructive report. In the manuscript, we have added and corrected the points you have mentioned. Below, you can see our answers to each point.

In the manuscript, red colored text denotes to added information, which weren’t in the first version and blue colored to a shift of phrase position without any further change.

1) The polymers were synthesized by radical polymerization of monomeric methacrylates. Mainly, the polymers 2 and 3 have advantageous properties because of their three-core side-chain and the cyano-substitution (Chapter “Materials and Methods”, line 101-103). Because of this, the property of our azo molecules results in a high anisotropy of the polarizability, large dipole moment and high birefringence (see line 98-99).

Since the synthesis and the physical properties are already explained in the references [24-26], we renounced a deeper description of it. However, we discussed the effect of those properties for an explanation of our results (Chapter “Discussion”, line 272-284). We think that the large polarizability, dipole moment and high birefringence explains the high azimuthal rotations we have measured. The bicyano-substituted, three-core side-chain of polymer 2 seems more advantageous to achieve higher azimuthal rotations than the monocyano-substituted one of polymer 3.

2) With the excitation wavelength of 473 nm, we address either side-chain of polymer 3. The two-core side-chain absorbs quantitatively a higher intensity since it is close to the center of its absorption band (line 119-121).  The excitation wavelength is positioned at the edge of the absorption band of the three-core side-chain. Compared to polymer 2, a reason for lower azimuthal rotation of P3 is that the two-core side-chain and the monocyano-substitution at the three-core side-chain is less advantageous and results in weaker physical properties mentioned in 1). These physical properties might be quantitively averaged for polymer 3 and, therefore, weaker. Based on references [24-26], where these properties of the polymers were discussed in detail, we think that the three-core side-chain contributes dominantly to the resulted photo-induced anisotropy and, consequently, to the quantity of the azimuthal rotation (see Chapter “Discussion”, line 272-284).

3) Unfortunately, we cannot follow which information from thickness dependent absorption spectra we can use for the discussion, besides increasing absorption for increasing thicknesses. We would be delighted to understand, if you have a special idea of which further information we can gain from thickness dependent absorption spectra that we didn’t consider. Please note that we didn’t discussed the penetration depth but the layer thickness.

However, it may help that the the absorption spectra were measured and displayed for various irradiation time in Ref. [24-26].

4) The effect of irradiation the sample from azo or glass-side is the main question of our previous work with the polymers and their character of light-induced chirality (see Ref. [27]). A brief discussion is added in Chapter “Discussion”, line 284-289.

5) The fluence of the used excitation laser is added in line 121-122 (Chapter “Materials and Methods”).

Reviewer 2 Report

The present paper shows photo-induced supramolecular chirality in thin films of achiral amorphous polymers with azo groups. I must admit that I am not an expert in the field. Anyway, the paper is well-written, and the results are clearly presented. Therefore, I would recommend publication of this paper.

Author Response

Dear Reviewer,

Thank you very much for your report and supporting our work. It is a great motivation for us.

Reviewer 3 Report

This manuscript describes chirality of achiral PMMA bearing azobenzene side chains induced by elliptically polarized light irradiation. Authors clarified not only the effect of the polymer film thickness on the azimuthal rotation and the ellipticity of the polarized light but also the temporal stability of the induced chirality. The obtained knowledges may be useful to understand chiral induction processes of polymer films. However, the manuscript is not suitable for publication in the journal “Molecules”. Because there was no chemical information such the molecular packing/conformation of the PMMA backbone and the trans-to-cis photoisomerization of the azobenzene side chains for relation to the induced chirality. I would suggest authors submit the manuscript in other journal focusing “polymer physics”.

Author Response

Dear Reviewer,

Thank you very much for your very constructive report. It helps us a lot to reflect our work. Since this manuscript is for a special edition on azo compounds, we feel it would fit for a collection of papers on the different aspects of azo materials.

In our research routine, the chemical information and the photo-isomerization of azo materials is getting so common and basic for us that we even didn’t think about it. Therefore, we gratefully appreciate for referencing to this. We added a description of the chemical information of photo-isomerization and visualized it by a figure (see line 23-33 and Fig. 1 on Page 1). Additionally, we tried to visualize the PMMA-backbone in Fig. 2b. Detailed information on the photochemistry and its mass flow behavior are available from the important work of Natansohn et al. in Ref [20] and Shibaev et al. in [28].

Round 2

Reviewer 1 Report

Authors almost addressed the comments from reviewers. However, polymer 3 seems to be different from the polymer in the Refs. [25-27] in that the three-core is monocyanosubstituted.

Reviewer 3 Report

Authors fully addressed the comments from reviewers. The revised manuscript will be accepted for publication in this journal without alterations.